# Vertigo in Patients with Degenerative Cervical Myelopathy

**DOI:** 10.3390/jcm10112496

**Published:** 2021-06-04

**Authors:** Zdenek Kadanka, Zdenek Kadanka, Rene Jura, Josef Bednarik

**Affiliations:** 1Department of Neurology, University Hospital, 625 00 Brno, Czech Republic; Kadanka.Zdenek@fnbrno.cz (Z.K.S.); Jura.Rene@fnbrno.cz (R.J.); Bednarik.Josef@fnbrno.cz (J.B.); 2Faculty of Medicine, Masaryk University, 625 00 Brno, Czech Republic

**Keywords:** cervical vertigo, cervical dizziness, degenerative cervical myelopathy, degenerative cervical spinal cord compression, cervical torsion test

## Abstract

(1) Background: Cervical vertigo (CV) represents a controversial entity, with a prevalence ranging from reported high frequency to negation of CV existence. (2) Objectives: To assess the prevalence and cause of vertigo in patients with a manifest form of severe cervical spondylosis–degenerative cervical myelopathy (DCM) with special focus on CV. (3) Methods: The study included 38 DCM patients. The presence and character of vertigo were explored with a dedicated questionnaire. The cervical torsion test was used to verify the role of neck proprioceptors, and ultrasound examinations of vertebral arteries to assess the role of arteriosclerotic stenotic changes as hypothetical mechanisms of CV. All patients with vertigo underwent a detailed diagnostic work-up to investigate the cause of vertigo. (4) Results: Symptoms of vertigo were described by 18 patients (47%). Causes of vertigo included: orthostatic dizziness in eight (22%), hypertension in five (14%), benign paroxysmal positional vertigo in four (11%) and psychogenic dizziness in one patient (3%). No patient responded positively to the cervical torsion test or showed significant stenosis of vertebral arteries. (5) Conclusions: Despite the high prevalence of vertigo in patients with DCM, the aetiology in all cases could be attributed to causes outside cervical spine and related nerve structures, thus confirming the assumption that CV is over-diagnosed.

## 1. Introduction

Dizziness and vertigo are among the most common complaints that lead patients to visit a physician. The lifetime prevalence in adults is around 20%, reaching 40% in older adults [1]. Vertigo is not a single disease entity but a symptom of a wide range of diseases of varying aetiology. These may arise from the inner ear, the brainstem, and the cerebellum, or they may be of internal, vestibular, or psychosomatic origin.

“Cervical (or cervicogenic) vertigo” (CV) is a term often used in a clinical practice, but physicians lack sufficient data to form definite opinions and to give clinical guidelines for its diagnosis and treatment [2]. The overall prevalence of CV is not known because there are no generally accepted clinical or paraclinical tests for CV, and therefore it is predominantly a diagnosis by exclusion [3]. Colledge et al., in a community-based sample of subjects over 65 years of age, found that cervical spondylosis is the second most frequent cause of dizziness [4]. Takahashi, in an out-patient sample of 1000 patients visiting a general hospital in Japan with a chief complaint of dizziness, estimated a prevalence of CV as high as 90% [5]. These data are in striking contrast to the opinion of several leading experts in the field who doubt the diagnosis of CV entirely [6].

Based on these findings and discrepancies, we hypothesised that CV is over-diagnosed due to the absence of detailed diagnostic theory and practice in papers that reported a high prevalence of CV. As degenerative cervical myelopathy (DCM) is the most severe symptomatic form of cervical spondylosis [7], we used a well-defined cohort of DCM patients to verify our hypothesis. The aim of this paper was to assess the prevalence and cause of vertigo in these patients with special focus on CV.

## 2. Materials and Methods

### 2.1. Design

This study was designed as a cross-sectional, cohort, observational, non-interventional study.

### 2.2. Participants

The study sample consisted of a cohort of consecutive subjects referred to a large tertiary university hospital between March 2018 and December 2019 in whom a clinical diagnosis of DCM was established, based on the presence of at least one clinical sign and one clinical symptom of myelopathy and magnetic resonance imaging (MRI) signs of degenerative discogenic and/or spondylogenic cervical spinal cord compression [8,9].

Excluded were:Patients with previous surgery on the cervical spine (possibly limiting the rotation of the spine);Patients with other than degenerative cervical cord compressions or other non-compressive myelopathies.

All subjects gave their written, informed consent to participate in the study.

### 2.3. Clinical Evaluation

Clinical neurological evaluation was focused on the assessment of clinical signs and symptoms of symptomatic myelopathy (with other possible causes excluded) and possible causes of vertigo. This included a detailed history of the illness, presence of comorbidities (cardiovascular including arterial hypertension, otorhinolaryngological and psychiatric abnormities, etc.), history of significant head or cervical spine trauma, Hallpike manoeuvre and a dedicated vertigo questionnaire (see below). 

The following symptoms and signs were sought and/or determined as markers of DCM:

Symptoms
Gait disturbance;Numb and/or clumsy hands;Lhermitte’s phenomenon;Bilateral arm paresthesias;Weakness of lower or upper extremities;Urinary urgency, or incontinence.

Signs
Corticospinal tract signs;Hyperreflexia/clonus;Spasticity;Pyramidal signs (Babinski reflex or Hoffman’s sign);Spastic paresis of any of the extremities (most frequently, lower spastic paraparesis);Flaccid paresis of one or two upper extremities;Atrophy of the hand muscles;Sensory involvement in various distributions in upper or lower extremities;Gait ataxia.

The following clinical and demographic data were also noted:Age;Sex.

Degree of disability was assessed by the modified Japanese Orthopaedic Association (mJOA) score [10]. 

### 2.4. Imaging

All subjects underwent examination of the cervical spine on a 1.5 Tesla MRI device with a 16-channel head and neck coil. The standardised imaging protocol included conventional pulse sequences in sagittal-T1, -T2 and STIR (short-tau inversion recovery) and axial planes (gradient-echo T2). The imaging criterion for cervical cord compression was defined as a change in spinal cord contour at the level of an intervertebral disc on axial or sagittal MRI scan compared with that at midpoint level of neighbouring vertebrae [11]. Compression ratio (CR) was calculated by taking the anterior–posterior diameter of the spinal cord divided by the transverse diameter of the cord on the axial image [11]. Lower CR values indicate worse cord deformation. This measurement was taken at the level of maximum spinal cord compression identified as maximum reduction in antero/posterior spinal canal diameter in comparison with other segments. The level of maximal spinal cord compression and signs of myelopathy (signal changes of the spinal cord on T1- and T2-weighted imaging) were also established (Figure 1).

### 2.5. Vertigo Questionnaire

Vertigo/dizziness was defined as an unpleasant disturbance of spatial orientation or to the erroneous perception of movement [12]. An investigator-administered questionnaire originally published by Filippopoulos was administered verbally to all patients [13]. The prevalence, the type of vertigo and the body positions and movements related to the different vertigo types were assessed by a series of questions. The questionnaire is shown in Appendix A. 

### 2.6. Uncontrolled Blood Pressure

Patients reporting any dizziness/vertigo were asked to measure their blood pressure at home under basal conditions 3 times daily for 3 consecutive days and the average value was then calculated. Uncontrolled blood pressure was defined as an average value ≥140 (systolic)/90 (diastolic) mm Hg. In borderline values 24-h monitoring of blood pressure was performed and the same definition was used for uncontrolled blood pressure.

### 2.7. Orthostatic Hypotension

Orthostatic hypotension was evaluated in patients reporting dizziness/vertigo by measuring blood pressure after lying flat for 5 min, then 1 min and 3 min after standing. For determination of orthostatic hypotension, we used an updated definition of the American Autonomic Society as a systolic blood pressure decrease of at least 20 mm Hg or a diastolic blood pressure decrease of at least 10 mm Hg within three minutes of standing when compared with blood pressure from the sitting or supine position [14].

### 2.8. Benign Paroxysmal Positional Vertigo 

Diagnostic criteria for benign paroxysmal positional vertigo (BPPV) consisted of vertical–torsional positional nystagmus evoked by the Dix–Hallpike manoeuvre or a predominantly horizontal positional nystagmus after rolling the head sideways from the supine position [15].

### 2.9. Ultrasound of Carotid and Vertebral Arteries

All ultrasound examinations were performed by an experienced neurosonologist using advanced ultrasound equipment (Philips PureWave HD 15; Massachusetts, USA) with a 3–12 MHz multi-frequency ultrasound probe. Patients were examined in a supine position with the neck slightly extended. Arterial wall thickness was evaluated and any extracranial atherosclerosis and/or occlusive disease was detected, with particular attention to the carotid bifurcation. In the event of carotid stenosis, its severity was measured in B mode and colour mode, with complementary measurements of peak systolic flow velocity and diastolic velocity gauged by Doppler ultrasound, based on the European Carotid Surgery Trial criteria (70–99% stenosis was considered significant) [16]. Vertebral arteries (VAs) were visualised in a longitudinal plane at the sixth cervical vertebra, where the vertebral artery usually enters the transverse foramina. For analysis, the course of the VA was divided into two segments: Vertebral (V1) (from the origin of the vertebral artery until the point where it enters the fifth or sixth cervical vertebra) and V2 (the part of the vertebral artery that courses cranially to the transverse foramina until it emerges besides the lateral mass of the atlas) [17]. Each segment of the VA was studied in B mode and colour-code mode. Any stenotic lesions of the VAs were evaluated according to B mode and flow pattern. Criteria used for grading ≥50% stenosis were focal elevated blood flow velocity with a PSV cut-off point at the V1 segment of the vertebral artery of 140 cm/s, and 125 cm/s at the V2 segment [18].

### 2.10. Cervical Torsion Test 

A cervical torsion test was performed in all patients. The procedure was adapted after the work of L’Hereux-Lebeau [19]. Subjects were seated in a rigid but fully rotatable chair that provided support to the entire body. Their legs were flexed with a slight bend at the knees. They were securely held in the chair with shoulder- and lap-belts. It was requested that their eyes should be closed during the procedure. First, the subject´s trunk was passively turned 70 degrees to the right, with the head still, then returned to centre, followed by turning the trunk 70 degrees to the left, and returning to centre. Each position was held for 30 s with the head stabilised by the observer for all positions. Nystagmus was evaluated with Frenzel goggles. The test was considered positive when nystagmus was found in any of the four positions, or vertigo provoked or increased (Figure 2).

The final diagnosis of DCM, together with the diagnosis of possible causes of vertigo, was defined by a neurologist (ZKJ) and then reviewed and confirmed by two other researchers (ZK and JB). Finally, detailed internal, otorhino-laryngological, neuro-otological or psychiatric examinations were performed according to suspected aetiology and the definite cause of vertigo was additionally verified by a highly qualified specialist. In case of discordance with the cause suspected by a neurologist, the final cause was established by consensus. We always cooperated with the same specialist.

## 3. Results

### 3.1. Study Cohort

We screened 51 patients in whom a diagnosis of DCM was established. Eight of them were excluded because of previous cervical spine surgery and five of them were not willing to participate in the study and did not sign informed consent. Thirty-eight patients complied with the DCM diagnosis and exclusion criteria, signed informed consent and completed the study protocol. The study cohort included 17 females (44.7%) with a median age (range) of 59 (41–85) years. The average mJOA score of the evaluated cohort was 16 (median), 9–17 (range). None of them reported significant injury of the head or cervical spine during the last year before inclusion in the study. Detailed demographic and imaging characteristics are summarised in Table 1. 

### 3.2. Dizziness/Vertigo

Subjective feelings of dizziness/vertigo in the previous six months were reported by 18 patients (47%). Patients characterised dizziness/vertigo as a feeling of impending blackout when rapidly standing up (eight patients), as a spinning vertigo (like in a carrousel) (five patients), as a swaying vertigo (like on a small boat) (four patients) and one patient was not able to specify it. Detailed characteristics of dizziness/vertigo and its aetiology in DCM patients are summarised in Table 2.

The following causes of vertigo were found in these patients: orthostatic dizziness in eight patients (44% of patients with vertigo, 22% of all patients), uncontrolled arterial hypertension in five (28% and 14%, respectively), BPPV in four (22% and 11%, respectively) and psychogenic dizziness in one (6% and 3%, respectively). The presence of uncontrolled arterial hypertension had to be confirmed by 24-h monitoring in two out of five patients (Table 2). 

None of the 38 patients studied displayed a positive response to the cervical torsion test, irrespective of the presence or absence of subjectively described vertigo in the previous six months. 

Three patients (0.8%) exhibited haemodynamically significant stenosis of the internal carotid arteries (two of them suffered from recently diagnosed, uncontrolled hypertension, while one had orthostatic dizziness). None of the patients studied had significant stenosis of the vertebral arteries (Table 1). 

## 4. Discussion 

Our study demonstrated a high prevalence of dizziness/vertigo in a cohort of patients with severe cervical spondylosis. Dizziness/vertigo was reported by 47% of the DCM patients. The aetiology of dizziness/vertigo in all patients in our DCM cohort, however, could be explained by mechanisms other than lesion(s) of the nervous system in the cervical region (i.e., orthostatic dizziness, uncontrolled hypertension, BPPV, psychogenic dizziness) or stenotic changes in the cervical segment of vertebral arteries. We thus have not been able to present any evidence in favour of the high prevalence of cervical dizziness/vertigo attributed either to advanced symptomatic spondylosis of the cervical spine and/or stenotic changes of vertebral arteries reported by other authors [4,5,20].

Vertigo, in general, is a common condition, yet definitions vary and management guidelines are often contradictory [21]. Patients with intrinsic problems (cardiovascular, pulmonary, etc.) are unlikely to suffer from pure rotational vertigo and the severity of this condition is often overrated by their clinicians [6]. Orthostatic dizziness in the adult population has accounted for 42% of all participants with vertigo and for 55% of non-vestibular dizziness diagnoses [22]. These findings correlate with the results of this study—in 44% of symptomatic (vertigo-suffering) patients, orthostatic aetiology was confirmed. Five patients were diagnosed with uncontrolled hypertension, making up 28% of the symptomatic group. In general, hypertension and dizziness are both highly prevalent and significantly associated, highlighting a pressing need for investments in preventive measures [23]. BPPV is the most common of the peripheral types of vertigo. Tan noted that 9% of elderly patients undergoing general geriatric assessment exhibited unrecognised BPPV [24]. This percentage proved even higher in a larger series of patients—approximately 34% [25]. Our study disclosed four patients with BBPV (22% of symptomatic subjects), but the group was too small to draw any definite conclusions. We decided to use the questionnaire by Filippopoulos to determine the prevalence of vertigo [13]. Unfortunately, the questionnaire cannot exactly differentiate between possible underlying pathologies, but it can lead us in a certain direction. A feeling of impending blackout when standing up rapidly is typical for orthostatic dizziness [26]. Vertigo (mostly spinning) triggered by head movements is typical for benign paroxysmal vertigo [27]. Swaying vertigo is often described as a somatoform and/or phobic vertigo [28]. In recent decades, cervical vertigo has emerged as a special category of dizziness, generating considerable controversy. The diagnosis remains debatable; there remains a lack of validated tests to confirm this entity, and exclusion clinical diagnosis appears to be the default standard [3,19]. A diagnosis of CV, however, is made too often by many physicians, largely because the simultaneous occurrence of vertigo and cervical spondylosis is very common [29]. Several explanations of the aetiology of CV have been published. Disturbed cervical proprioception is suggested by probably the most cited study [30]. Neck afferents (nerves) not only assist the coordination of eye, head, and body, but they also affect spatial orientation and control of posture. This implies the theory that stimulation of, or lesions (damage) in, these structures could produce CV [31]. In experimental studies, vertigo, ataxia, and nystagmus have been induced in animals by injecting local anaesthetics into the neck [32]. Ataxia in healthy human beings, induced by unilateral injection of local anaesthetics in the neck, has also been associated with a broad-based, staggering gait and hypotonia of the ipsilateral arm and leg [32]. According to these findings, some authors have suggested that cervical spinal cord compression is the most frequent cause of cervical vertigo [33]. The cervical torsion test is supposed to be the most useful to distinguish between cervical afferent disturbance and vestibular dysfunction in patients with dizziness/vertigo [19]. It is the reason why it was used in this study to elucidate the role of the cervical proprioceptors in DCM patients. The principle of the test is to achieve stimulation of the proprioceptors of the neck; the trunk of the body is rotated with the head kept stationary. This examination, however, was not able to evoke vertigo in any patient in this cohort. The second most common hypothesis as to the aetiology of CV is that it may arise out of impaired blood circulation in the vertebrobasilar arteries. In 1933, DeKleyn first described a syndrome of vertigo produced by head movement. In post-mortem studies, he noted compromised circulation in the VA with head rotation. Later, stroke accompanying maximum rotation of the head was described in archery [30]. However, because of the collateral blood flow through the contralateral VA and the circle of Willis, VA occlusion does not lead to symptoms in most individual cases. Thus, cases of symptomatic rotational vertebral artery occlusion are very rare [34]. Investigation of the effect of the position of the head on flow rate in the vertebral arteries, as measured by Doppler ultrasound at rotations of 30 degrees up to 60 degrees to either side, revealed no changes in blood flow in healthy subjects, which means that common rotation of the cervical spine cannot elicit vertigo [35]. Thus, in conclusion, the available literature indicates that hypoperfusion in the vertebrobasilar territory has no close correlation with clinical symptoms of cervical vertigo, and should not be raised as the sole reason in explaining CV [36,37]. This finding was also confirmed by this study. Moreover, vertebrobasilar insufficiency remains a controversial clinical entity lacking clear diagnostic criteria [38]. 

### Limitations of the Study 

This study has several limitations. The sample size is small. However, we consider a cohort of 30–40 DCM patients large enough to confirm the hypothesis of CV as a prevalent condition; we used robust inclusion/exclusion criteria and an extensive evaluation, including neurological and vestibular clinical assessments. Our results have limited importance only for patients with severe cervical spondylosis and symptomatic cervical myelopathy, not for other conditions or a general population. We used the cervical torsion test to evaluate the role of cervical proprioceptors in the pathophysiology of CV, but there are no generally accepted clinical or paraclinical tests for CV and therefore it is predominantly a diagnosis by exclusion.

## 5. Conclusions

In conclusion, despite a comparatively high prevalence (47%) of dizziness/vertigo in patients with severe cervical spondylosis, it is primarily necessary to be in doubt about the diagnosis of so-called “cervical vertigo” and to seek other (often treatable) aetiologies, thus avoiding the possibility of overlooking other serious neurological, otorhinolaryngological or circulatory problems.

## Figures and Tables

**Figure 1 jcm-10-02496-f001:**
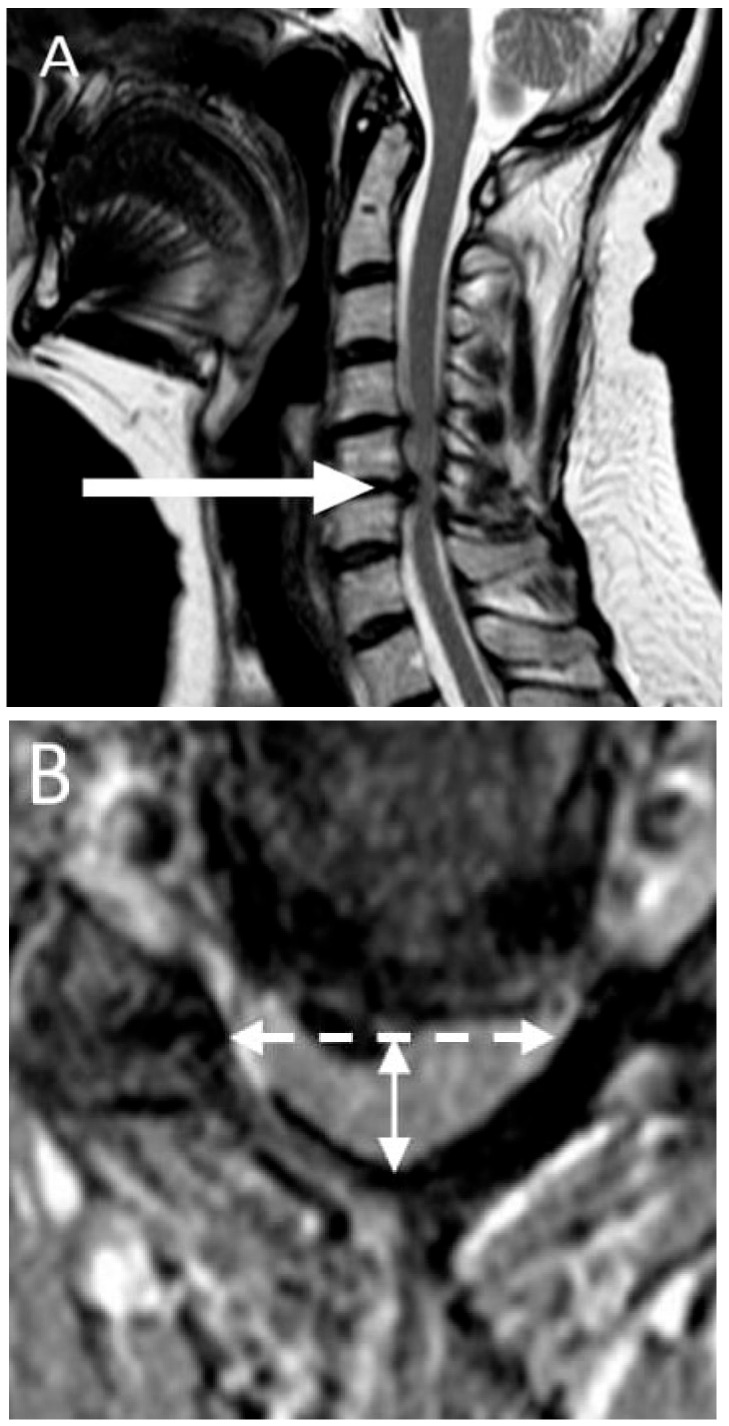
Patient with severe cervical spinal cord compression. (**A**) Sagittal T2-MRI sequence shows a level of maximal compression—C5/6 (arrow); (**B**) Compression ratio: anterior–posterior diameter (solid line double arrow) divided by the transverse diameter (dashed line double arrow) of the spinal cord on the axial T2 MRI image (taken at the level of maximum spinal cord compression; the result is 0.37 in this patient).

**Figure 2 jcm-10-02496-f002:**
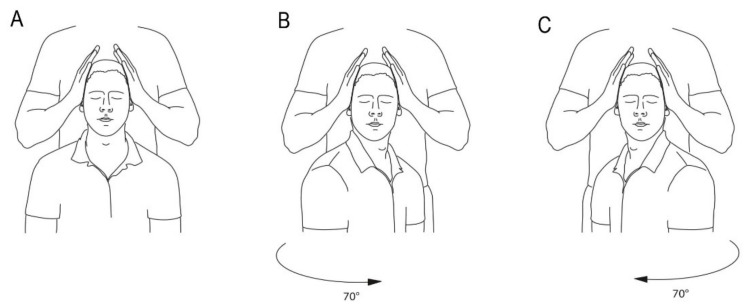
Cervical torsion test. (**A**) Subject seated in a rigid but fully rotatable chair, head fixed. (**B**) The subject´s trunk passively turned 70 degrees to the right, with the head still, then returned to centre. (**C**) Turning the trunk 70 degrees to the left, then returned to centre.

**Table 1 jcm-10-02496-t001:** Demographic and imaging characteristics.

Patients No	Gender	Age	mJOA Score	Maximum Cervical Cord Compression Level	Signs of Myelopathy on MRI	CR	ICA Stenosis	VA Stenosis
1	F	46	17	C5/6	no	0.31	no	none
2	M	44	17	C4/5	no	0.3	no	none
3	F	60	17	C6/7	yes	0.32	no	none
4	F	60	16	C6/7	no	0.37	no	none
5	F	51	17	C4/5	no	0.44	no	none
6	M	43	16	C4/5	no	0.45	no	none
7	M	71	15	C5/6	yes	0.35	yes	none
8	M	60	17	C4/5	no	0.3	no	none
9	M	65	16	C3/4	yes	0.4	no	none
10	M	51	16	C5/6	yes	0.43	no	none
11	M	65	17	C5/6	no	0.36	no	none
12	F	50	17	C5/6	no	0.36	no	none
13	F	63	11	C5/6	yes	0.36	yes	none
14	F	71	16	C5/6	no	0.41	no	none
15	M	58	16	C4/5	no	0.43	no	none
16	F	69	12	C4/5	yes	0.39	no	none
17	M	60	15	C6/7	yes	0.42	no	none
18	F	59	16	C6/7	no	0.40	no	none
19	M	63	17	C5/6	yes	0.42	no	none
20	M	52	16	C5/6	no	0.28	no	none
21	M	69	15	C5/6	no	0.3	no	none
22	F	57	16	C5/6	yes	0.38	no	none
23	M	82	17	C6/7	no	0.36	no	none
24	F	59	15	C5/6	yes	0.36	no	none
25	M	67	13	C5/6	yes	0.49	yes	none
26	M	64	15	C5/6	no	0.41	no	none
27	M	45	17	C3/4	no	0.37	no	none
28	M	77	9	C4/5	yes	0.41	no	none
29	F	40	17	C5/6	no	0.43	no	none
30	M	59	17	C5/6	yes	0.44	no	none
31	F	51	17	C5/6	no	0.39	no	none
32	M	48	15	C5/6	yes	0.21	no	none
33	F	48	17	C5/6	no	0.23	no	none
34	F	59	17	C5/6	no	0.33	no	none
35	F	48	17	C4/5	yes	0.23	no	none
36	F	52	17	C4/5	no	0.44	no	none
37	M	58	16	C5/6	yes	0.38	no	none
38	M	68	17	C5/6	no	0.39	no	none

mJOA: modified Japanese Orthopaedic Association score; CR: compression ratio; ICA: internal carotid artery; VA: vertebral artery; MRI: magnetic resonance imaging; F: female; M: male.

**Table 2 jcm-10-02496-t002:** Detailed characteristics of dizziness/vertigo and its aetiology in DCM patients.

Patients No	Type of Vertigo	Vertigo According to Body Movement	Cervical Torsion Test	Hallpike Test	Drop in BP ≥ 20/10 mmHg after at Least 3 min of Standing	Upright Tilt Table Test	Uncontrolled AH Detection	Final Aetiology of Dizziness
1	none	none	negative	negative	No	NA		NA
2	none	none	negative	negative	No	NA		NA
3	none	none	negative	negative	No	NA		NA
4	none	none	negative	negative	No	NA		NA
5	unspecified dizziness	also present when sitting or lying down	negative	negative	No	NA	24 h monitoring	uncontrolled AH
6	none	none	negative	negative	No	NA		NA
7	spinning	also present when sitting or lying down	negative	negative	No	NA	self-measurement	uncontrolled AH
8	none	none	negative	negative	No	NA		NA
9	none	none	negative	negative	No	NA		NA
10	blackout when standing	triggered by a change of position	negative	negative	Yes	NA		orthostatic vertigo
11	none	none	negative	negative	No	NA		NA
12	blackout when standing	triggered by a change of position	negative	negative	No	positive		orthostatic vertigo
13	swaying	only present when standing or walking	negative	negative	No	NA	24 h monitoring	uncontrolled AH
14	blackout when standing	triggered by a change of position	negative	negative	Yes	NA		orthostatic vertigo
15	none	none	negative	negative	No	NA		NA
16	swaying	also present when sitting or lying down	negative	negative	No	NA		psychogenic vertigo
17	none	none	negative	negative	No	NA		NA
18	swaying	only present when standing or walking	negative	negative	No	NA	self-measurement	uncontrolled AH
19	none	none	negative	negative	No	NA		NA
20	none	none	negative	negative	No	NA		NA
21	blackout when standing	triggered by a change of position	negative	negative	Yes	NA		orthostatic vertigo
22	none	none	negative	positive	No	NA		NA
23	blackout when standing	triggered by a change of position	negative	negative	Yes	NA		orthostatic vertigo
24	spinning	triggered by head movement	negative	positive	No	NA		BPPV
25	blackout when standing	triggered by a change of position	negative	negative	Yes	NA		orthostatic vertigo
26	blackout when standing	triggered by a change of position	negative	negative	Yes	NA		orthostatic vertigo
27	none	none	negative	negative	No	NA		NA
28	spinning	trigered by head movement	negative	positive	No	NA		BPPV
29	none	none	negative	negative	No	NA		NA
30	none	none	negative	negative	No	NA		NA
31	spinning	triggered by head movement	negative	positive	No	NA		BPPV
32	blackout when standing	trigered by a change of position	negative	negative	No	positive		orthostatic vertigo
33	spinning	triggered by head movement	negative	positive	No	NA		BPPV
34	swaying	only present when standing or walking	negative	negative	No	NA	self-measurement	uncompensated AH
35	none	none	negative	negative	No	NA		NA
36	none	none	negative	negative	No	NA		NA
37	none	none	negative	negative	No	NA		NA
38	none	none	negative	negative	No	NA		NA

NA: not attributable; BPPV: benign paroxysmal positional vertigo; AH: arterial hypertension; BP: blood pressure.

## Data Availability

The data presented in this study are available on request from the corresponding author. The data are not publicly available.

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
