# Peer review of "Vertigo in Patients with Degenerative Cervical Myelopathy"

_jcm, 2021, doi:10.3390/jcm10112496_

Round 1

Reviewer 1 Report

Dear Authors,

first of all, I thank You for giving me the opportunity to read this Your manuscript, submitted for publication.

I read it with interest.

I have only minor comments and suggestions. I hope that they can be useful for You.

a) is the cerebellum a part of the brainsistem ? (please, read linesc 29-30)

b) otorhinolaryngogical (line 66) : not in italics

c) (more than 20 years) (line 69). Please remove because not useful and not "elegant"

d) reference no. 11 was unclear. Please, clarify according to journal instructions

Author Response

Dear Authors, first of all, I thank You for giving me the opportunity to read this Your manuscript, submitted for publication. I read it with interest. I have only minor comments and suggestions. I hope that they can be useful for You.

Comment 1: Is the cerebellum a part of the brain stem? (please, read lines c 29-30)

Response: No, the cerebellum is not a part of the brain stem. Corrected in the manuscript, to be more understandable

Comment 2: otorhinolaryngogical (line 66) : not in italics

Response: corrected

Comment 3: (more than 20 years) (line 69). Please remove because not useful and not "elegant"

Response: removed

Comment 4: reference no. 11 was unclear. Please, clarify according to journal instructions

Response: corrected

Reviewer 2 Report

Thank you for asking me to review this paper exploring the causes of dizziness across a cohort of 38 patients with degenerative cervical myelopathy. 

This is a well-conducted study. The sample size is relatively small given but there are robust inclusion/exclusion criteria, and patients have undergone an extensive evaluation, including neurological and vestibular clinical assessments. 

Some minor comments:

Methods

Dizziness questionnaires explored symptoms of dizziness across a 6-month period. How many patients had ongoing dizziness? 

Was there an attempt to quantify the severity of the vertigo?

I would have liked to have seen a figure depicting the cervical torsion test, as this is not a standard test that many readers would be familiar with (and the rationale for including this is only found late in the discussion).

An approach to correlating dizziness/vertigo severity with spondylosis severity would have been interesting (as one would have expected these to de-correlate).

Results

I was interested in the finding of uncontrolled hypertension in 28% of the symptomatic group. What was the rationale for considering this assessment? How do the authors link raised blood pressure with dizziness? This may be something to explore further in the discussion.

Discussion:

Only 4 patients had BPPV from this cohort and the authors correctly identify that the small sample size may explain the relatively low rates, although the median age range of 59 may also account for this.

The authors discuss the (also misleading) concept of vertebrobasiar insufficiency as an explanation for cervicogenic vertigo. There is a recent review of this in Practical Neurology (Chandratheva et al., 2021) that may be of interest to the authors.

Author Response

Thank you for asking me to review this paper exploring the causes of dizziness across a cohort of 38 patients with degenerative cervical myelopathy. This is a well-conducted study. The sample size is relatively small given but there are robust inclusion/exclusion criteria, and patients have undergone an extensive evaluation, including neurological and vestibular clinical assessments.  Some minor comments:                                                               Methods:

Comment 1: Dizziness questionnaires explored symptoms of dizziness across a 6-month period. How many patients had ongoing dizziness? 

Response: Most patients (22) had repeated shorts episodes of vertigo.

Comment 2: Was there an attempt to quantify the severity of the vertigo?

Response: No, we have just tried to determine the prevalence, type (character) and cause of vertigo in DCM patients. There are some scores that quantify the severity of vertigo like “Vertigo symptom scale”, but we have decided to use other questionnaire. Moreover, the severity of spondylosis does not often correlate with the severity of myelopathic symptoms (see the response to Comment 4)

Comment 3: I would have liked to have seen a figure depicting the cervical torsion test, as this is not a standard test that many readers would be familiar with (and the rationale for including this is only found late in the discussion).

Response: A figure was added to the manuscript

Comment 4: An approach to correlating dizziness/vertigo severity with spondylosis severity would have been interesting (as one would have expected these to de-correlate).

Response: It is really an interesting idea. However, it was not the aim of this paper, because severity of spondylosis (or even severity of degenerative compression of the spinal cord) doesn’t often correlate with severity of myelopathic signs and symptoms (Tempest-Mitchell J, Hilton B, Davies BM, et al. A comparison of radiological descriptions of spinal cord compression with quantitative measures, and their role in non-specialist clinical management. PloS one 2019;14:e0219380. doi:10.1371/journal.pone.0219380      Nouri A, Tetreault L, Côté P, et al. Does Magnetic Resonance Imaging Improve the Predictive Performance of a Validated Clinical Prediction Rule Developed to Evaluate Surgical Outcome in Patients With Degenerative Cervical Myelopathy? Spine 2015;40:1092–100. doi:10.1097/brs.0000000000000919     Wilson JR, Barry S, Fischer DJ, et al. Frequency, timing, and predictors of neurological dysfunction in the nonmyelopathic patient with cervical spinal cord compression, canal stenosis, and/or ossification of the posterior longitudinal ligament. Spine 2013;38:S37-54.  doi:10.1097/brs.0b013e3182a7f2e7       Badhiwala JH, Witiw CD, Nassiri F, et al. Patient phenotypes associated with outcome following surgery for mild degenerative cervical myelopathy: a principal component regression analysis. Spine J 2018;18:2220–31. doi:10.1016/j.spinee.2018.05.009). Furthermore, we did not quantify severity of dizziness/vertigo (see response above).

Comment 5: Results: I was interested in the finding of uncontrolled hypertension in 28% of the symptomatic group. What was the rationale for considering this assessment? How do the authors link raised blood pressure with dizziness? This may be something to explore further in the discussion.¨

Response: Hypertension and dizziness are both highly prevalent and significantly associated (as mentioned in the “Discussion”) (Lopes, A. R.; Moreira, M. D.; Trelha, C. S.; Marchiori, L. L. de M. Association between Complaints of Dizziness and Hypertension in Non-Institutionalized Elders. Int. Arch. Otorhinolaryngol., 2013, 17 (2), 157–162.  M Middeke 1, B Lemmer, B Schaaf, L Eckes. Prevalence of hypertension-attributed symptoms in routine clinical practice: a general practitioners-based study. J Hum Hypertens. 2008 Apr;22(4):252-8. etc.).  We had a lot of patients in our clinical neurological practice, who were investigated for “a vertigo of unknown origin” and finally the uncompensated hypertension was found to be the cause of this problems.

Comment 6: Discussion: Only 4 patients had BPPV from this cohort and the authors correctly identify that the small sample size may explain the relatively low rates, although the median age range of 59 may also account for this.

Response: Yes, that is correct. The small sample size and age of patients was probably responsible for low percentage of BPPV in our cohort.

Comment 7: The authors discuss the (also misleading) concept of vertebrobasiar insufficiency as an explanation for cervicogenic vertigo. There is a recent review of this in Practical Neurology (Chandratheva et al., 2021) that may be of interest to the authors.

Response: Thank You for recommendation of this article. Added to discussion and citations.

Reviewer 3 Report

Dear Authors!

This is an interesting article dealing with the prevalence and diagnostic of cervical vertigo. Authors performed a prospective observational study including 38 DCM patients. 47% presented with vertigo. However, none cervical vertigo was found.

The article has several major and minor concerns limit the quality of the article.

Major concerns:

  1. Only 38 patients were included and analysed. Why did you not include 50 patients or 100 patients to improve the statistical weight of you study? (Even though you did not find any patient with cervical vertigo).
  2. Paragraph describing the “Vertigo questionnaire” is boring and should be shortened. English version of the questionnaire could be uploaded as supplementary data for interested readers.
  3. “Finally, detailed internal, otorhino-laryngological, neuro-otological or psychiatric examinations were performed according to suspected aetiology and the definite cause of vertigo was additionally verified by a highly qualified specialist. In case of discordance with the cause suspected by a neurologist, the final cause was established by consensus.” à was it the same specialist in all cases?
  4. Dizziness/vertigo should be clearly defined in the methods.
  5. Results/Table: The occurrence of a sign of myelopathy on T2-weighted and T1-weighted MRI should be added within the results and within table 1.
  6. Figure of a DCM patient should be added for illustration showing T1- and T2-weighted MRI and its maximum cervical cord compression level and its resulting Cr.

Minor concerns:

  1. Why did you exclude patients with previous cervical spine surgeries? Your thoughts should be shared within the methods.
  2. In my opinion, mJOA score should only be cited. As you wrote, it is generally accepted and known. Interested readers who do not know the mJOA Score can read the original article.
  3. Self-citation could be reduced.
  4. Sub-headings, especially all of 2.4, should be revised à eg. 2.4. Diagnostics, 2.4.1. Vertigo questionnaire etc,)
  5. Paragraph “2.4. Vertigo questionnaire” also describes blood pressure measurements, orthostatic hypotension and benign paroxysmal positional vertigo. This paragraph should be divided into different paragraphs.
  6. Line 68: “It was performed by a neurologist experienced (more than 20 years) in the management of myelopathic cases (ZKJ) --> should be removed. Does not add any additional information to the readers.
  7. Line 172: “…both highly experienced (more than 35 years) in clinical studies of myelopathy and neurology.” --> should be removed. Does not add any additional information to the readers.
  8. Line 185: “Age and sex characteristics of the 13 dropout patients were similar to subjects included in the study: age 61 (median); 47-76 years (range); 6 females 186 (46.1%).” à should be removed.
  9. Summary/paragraph of all abbreviations is missing

Overall, I would recommend "major revision" to improve the quality of the article.

Author Response

Major concerns:

Comment 1: Only 38 patients were included and analysed. Why did you not include 50 patients or 100 patients to improve the statistical weight of you study? (Even though you did not find any patient with cervical vertigo).

Response:  The aim of this paper was to determine the prevalence and cause of vertigo in DCM patients with special focus on “cervical vertigo”. Regarding referred prevalence of cervical vertigo (as some authors suppose that CV is the second most frequent cause of dizziness, other authors estimate cervical aetiology in 90% of inpatients in general hospital with a chief complaint of dizziness), we consider a cohort of 30-40 DCM patients large enough (using robust inclusion/exclusion criteria and an extensive evaluation, including neurological and vestibular clinical assessments) to confirm the hypothesis CV as a prevalent condition.

Comment 2: Paragraph describing the “Vertigo questionnaire” is boring and should be shortened. English version of the questionnaire could be uploaded as supplementary data for interested readers.

Response: This paragraph was shortened, and the English version of the questionnaire could was uploaded as supplementary data

Comment 3: “Finally, detailed internal, otorhino-laryngological, neuro-otological or psychiatric examinations were performed according to suspected aetiology and the definite cause of vertigo was additionally verified by a highly qualified specialist. In case of discordance with the cause suspected by a neurologist, the final cause was established by consensus.” à was it the same specialist in all cases?

Response: The discordance was extremely rare, and the consensus was made between investigating neurologist and the specialist. We always cooperated with the same specialist.

Comment 4: Dizziness/vertigo should be clearly defined in the methods.

Response: The definition was added to the methodology

Comment 5: Results/Table: The occurrence of a sign of myelopathy on T2-weighted and T1-weighted MRI should be added within the results and within table 1.

Response The occurrence of signs of myelopathy on T2/T1- weighted MRI was added to the table 1.

However, the prevalence of hyperintense signal on T2WI among patients with clinically confirmed DCM has been reported within the range of 58%–85% (Nouri et al 2016). It means, that a lot of patients with clinical DCM have no signs of myelopathy on MRI.

T2W hyperintensities may be non-specific and can be found even in patients without DCM signs (Bednarik et al. 2008). Hypointensities on T1WI are quite rare and mostly visible in severe myelopathies (Nouri et al 2016)

Comment 6: Figure of a DCM patient should be added for illustration showing T1- and T2-weighted MRI and its maximum cervical cord compression level and its resulting Cr.

Response: Figure was added to the manuscript.

Minor concerns:

Comment 7. Why did you exclude patients with previous cervical spine surgeries? Your thoughts should be shared within the methods.

Response: We have excluded them, because the performing of rotational cervical spinal test could be complicated in these patients. Moreover some authors published a theory, that the cervical spine operation can interrupt the “input of cervical afferents” and relieve the symptoms of cervical vertigo (B. Peng, L. Yang, C. Yang, X. Pang, X. Chen, Y. Wu. The effectiveness of anterior cervical decompression and fusion for the relief of dizziness in patients with cervical spondylosis a multicentre prospective cohort study. The Bone & Joint Journal.  Vol. 100-B, No. 1).

Comment 8. In my opinion, mJOA score should only be cited. As you wrote, it is generally accepted and known. Interested readers who do not know the mJOA Score can read the original article.

Response: Corrected, and mJOA score was only cited

Comment 9. Self-citation could be reduced.

Response: Two self-citations were deleted

Comment 10: Sub-headings, especially all of 2.4, should be revised à eg. 2.4. Diagnostics, 2.4.1. Vertigo questionnaire etc,)

Response: corrected

Comment 11: Paragraph “2.4. Vertigo questionnaire” also describes blood pressure measurements, orthostatic hypotension and benign paroxysmal positional vertigo. This paragraph should be divided into different paragraphs.

Response: corrected

Comment 12: Line 68: “It was performed by a neurologist experienced (more than 20 years) in the management of myelopathic cases (ZKJ) --> should be removed. Does not add any additional information to the readers.

Response: deleted

Comment 13:  Line 172: “…both highly experienced (more than 35 years) in clinical studies of myelopathy and neurology.” --> should be removed. Does not add any additional information to the readers.¨

Response: removed

Comment 14: Line 185: “Age and sex characteristics of the 13 dropout patients were similar to subjects included in the study: age 61 (median); 47-76 years (range); 6 females 186 (46.1%).” à should be removed.

Response: removed

 Comment 15: Summary/paragraph of all abbreviations is missing

Response: Paragraph of all abbreviations was not demanded by the editor. It is not mentioned in “Instruction for authors” as well. We can add it, if it will be required by the Editor.

Round 2

Reviewer 3 Report

Dear Authors,

Thank you very much for implementing the major and minor concerns.

Small sample size shlould be mentioned within the limitations despite the "robust" statistics.

At last I would like you to ask you to improve the quality of Fig. 1. --> Fig. 1 A could be focused by cutting/deliting the spine Sub C7. Fig 1 B should be centered and focused

Thank you very much!

Author Response

Comment 1: Small sample size shlould be mentioned within the limitations despite the "robust" statistics.

Response: It was added to the limitations of the study

Comment 2: At last I would like you to ask you to improve the quality of Fig. 1. --> Fig. 1 A could be focused by cutting/deliting the spine Sub C7. Fig 1 B should be centered and focused

Response: The Fig. 1 completely improved